# Development of a new high-yield integration site assay reveals disease-specific patterns across HTLV-1-associated pathologies

Vincent Guiraud,[1] Jérôme Alexandre Denis,[2] Sofia Ben Attia,[1] Erwan Ablin,[1] Ronan Legrand,[2] Véronique Morel,[3] Claire Lacan,[3] Sylvain Choquet,[3] Rabab Debs,[4] Margaux Cheval,[4] Cindy Marques,[5] Alexandre Le Joncour,[5] Jean-Christophe Corvol,[4] Olivier Benveniste,[5] Valérie Pourcher,[6,7] Anne-Geneviève Marcelin,[1] Agnès Gautheret-Dejean,[8,9] Clotilde Bravetti,[10] Vincent Calvez[1]

**ABSTRACT** Human T lymphotropic virus type 1 (HTLV-1) chronic infection is maintained through mitotic proliferation of the infected CD4+ T cells, where the viral genome is integrated as a provirus in its host genome. HTLV-1 integration sites (ISs) have a part in HTLV-1-associated pathologies, with distinct IS patterns associated with malignant proliferation or inflammatory diseases. However, IS determination remains challenging because most assays rely on complex biological and biocomputing protocols. We present an IS assay that solely relies on PCR and Sanger sequencing, which allowed HTLV-1 IS determination in four patients with various HTLV-1-associated pathologies. We adapted an IS assay derived from a panhandle PCR, with several modifications to increase yield. Absence of bias regarding IS retrieval was confirmed using TCRγ clonality. IS analysis was performed in four HTLV-1-positive patients: two with polymyositis, one with adult T-cell leukemia/lymphoma (ATLL), and one with HTLV-1-associated myelopathy (HAM). Overall yield was around 20%, with a mean sequence length downstream the IS of 336 ± 230 bp (range, 44–1,024 bp). There was no major bias in clonal determination, as IS results matched clonality assessed using a TCRγ assay. The IS assay revealed distinct clonal patterns depending on HTLV-1 pathology: dominated by a large clone for ATLL, oligo- or polyclonal for polymyositis and polyclonal for HAM. As a conclusion, we present an easy-to-implement integration site assay for HTLV-1 that allows a relatively unbiased IS analysis regarding clonal populations. This assay could be useful to further explore IS involvement in HTLV-1 associated pathologies.

**IMPORTANCE** Human T lymphotropic virus type 1 (HTLV-1) chronic infection is due to the mitotic proliferation of infected CD4+ T cells, where the proviral DNA is integrated in its host DNA. HTLV-1 integration seems to play a non-negligible part in HTLV-1-associated pathologies. However, most HTLV-1 integration studies originate from a few centers, mostly because integration site (IS) protocols rely on high-cost experimental materials and advanced bioinformatic analysis. We have developed an IS assay that solely relies on Taq polymerase and Sanger sequencing, with no need for costly biological material nor complex bioinformatic skills. This assay was successfully performed on four HTLV-1-positive patients with distinct pathologies (ATLL, HAM, and polymyositis) and distinct material (blood and cerebrospinal fluid). All four patients originated from distinct areas in Africa and the Caribbean Sea Island. This assay has a relatively high yield, around 20%. It provided similar results regarding HTLV-1 clonality compared with a TCRγ assessment, which indicated that IS recovery was likely unbiased.

**KEYWORDS** HTLV-1, Integration Site, ISLA, panhandle PCR, clonal proliferation

Address correspondence to Vincent Guiraud, vincent.guiraud@aphp.fr.

The authors declare no conflict of interest.

See the funding table on p. 5.

Human T lymphotropic virus type 1 (HTLV-1), the first human retrovirus discovered, affects 10 million people worldwide (1, 2). Around 5% of HTLV-1 carriers develop either an aggressive hematologic malignancy known as adult T-cell leukemia/lymphoma (ATLL) or an inflammatory condition, such as HTLV-1-associated myelopathy (HAM) or polymyositis (3, 4).

HTLV-1 chronic infection is sustained through mitotic proliferation of infected CD4+ T cells, where the viral genome persists as a provirus integrated in its host genome. The viral integration site (IS) within the host genome appears significant; several studies have identified differing IS patterns in asymptomatic carriers versus patients with ATLL or HAM (3, 5–8). However, determining HTLV-1 integration sites is technically challenging (9).

We present a high-yield HTLV-1 integration site (IS) assay using PCR and Sanger sequencing alone.

## MATERIALS AND METHODS

### Patients

All patients with an HTLV-1 diagnosis performed in Pitié-Salpêtrière Hospital and available samples collected for routine standard clinical management were included.

### HTLV quantification

Genomic (including HTLV-proviral) DNA was extracted from whole blood or cerebrospinal fluid (using a QiaSymphony (Qiagen) platform. Quantification was performed with previously published primers (10), using the Qx200 digital droplet platform (Bio-Rad). Results were normalized with the quantification of the albumin gene (11).

### Integration primer design:

HTLV-1 sequences were retrieved from GenBank with keywords "HTLV-1 LTR" and "complete genome HTLV-1" (12). Sequences were aligned using MAFFT (13) and visualized with UGENE (14). Primers were designed to have a melting temperature (Tm) of 60°C, secondary structures were screened with OligoCalc (15), and primer specificity was assessed using Primer-BLAST (16).

### Integration site assay

We used a modified panhandle PCR to capture HTLV-1 3′ integration sites with several modifications to increase yield (12, 13). First, genomic DNA was diluted into 96-well plates with two proviruses per well. Then, unidirectional linear extension was performed using 300 nM HTLV-1 primer (ACAGCCTGGCAAAACGGCCTCCTTCC), 1× GXL Buffer, 200 mM dNTP, and 5% DMSO in 30 µL. Cycling was 30 cycles (98°C 10 s, 60°C 15 s, and 68°C 3 min), followed by a progressive cooling down to 25°C (0.2 °C/s ramp) to promote genomic DNA renaturation. This last step is critical as it allows genomic DNA to hybridize (14), with added DMSO to accelerate the process (15). Then, a mixture of 5 IU Taq DNA polymerase (New England Biolabs, #M0273E), 1.5× Taq Buffer, 8 µM decaHTLV1.U5 (ACGGCCAAGTRCCGGCGACTNNNNNNNNNN) and 200 mM dNTP in 10 µL was added to each well. The reaction was incubated at 68°C for 2 min, 65°C for 1 min, cooled by 1°C per minute until 25°C, heated to 60°C, and ramped down by 1°C per minute until 20°C. This primer was in large excess to hybridize with the newly synthesized single-strand DNA, and Taq was added to polymerize the reverse complementary strand. Then, 10 IU of Exonuclease 1 (New England Biolabs, #M0293L) was added to the 40 µL reaction mix, with the following cycling: 45 min 37°C, heated by 1°C every 3 min until reaching 43°C, heated at 68°C for 15 min, 80°C for 15 min before cooling at 8°C. This step was performed to increase yield by removing previous primers and trimming ends to allow a unique sticky end that is complementary to the HTLV-I LTR. Then, loop formation was performed using 20 µL of the previous round with a fresh mix composed of 3.75 IU Taq,

1X Taq buffer, 5% DMSO, 200 mM dNTP, and 1 µM HTLV.RF2 (CACCCCTTTCCCTTTCATT CACGAC) in 30 µL. Cycling: 1 min at 95°C, 10 cycles (94°C for 20 s, 60°C for 30 s, 68°C for 2 min), 40 cycles (92°C for 10 s, 65°C for 15 s, and 68°C for 2 min), 68°C for 5 min. Using a fresh mix is critical as a long contact between DMSO and Taq inhibits the enzyme (16). The first 10 rounds were performed to favor intra-strand hybridization and complete the panhandle, while the following allows first amplifications. Finally, two nested PCRs were performed, using first HTLV.RF1 (CGACTRACTGCCGGCT) then HTLV1.U5 (ACGGC-CAAGTRCCGGCGACT). PCR mixtures were identical: 2 µL of previous round with 3.75 IU Taq, 200 mM dNTP, 0.6 µM Primer in 50 µL. Cycling was identical for both rounds: 95°C for 1 min, 35 cycles (94°C for 20 s, 65°C for 30 s, and 68°C for 2 min), 68°C for 5 min. Amplicons were visualized on a 1% agarose gel, with positive wells exhibiting a 1–5 kb amplicon (Fig. S1). Amplicons were purified and Sanger-sequenced using the (Fig. S1) HTLV.2.U5 primer (CCGTTGGCTCGGAGCCAG), and resulting sequences were analyzed using the "Integration Site" web tool as previously described (17). Consensus viral sequence adjacent to integration site is "AGTACACA."

## TCR gamma

TCR gamma rearrangement determination was performed using established methods (18).

## Statistics analyses

Figures and statistical analyses were performed using R or GraphPad Prism. Yield was calculated by dividing the total number of integration sites retrieved (after Sanger sequencing) by the total number of integration sites used as input.

## RESULTS

The integration site assay (IS) was performed in four HTLV-1-seropositive patients: two with polymyositis, one with ATLL, and one with HAM (Table 1). Blood samples were used for all patients except for P2 (HAM), for whom cerebrospinal fluid was analyzed. End-point dilutions, performed using ddPCR, were of two proviruses per well, except for P2 (1.2 provirus/well) because of the extremely limited number of proviruses available (Table S2). Considering all proviruses as an input, overall yield was around 20%: 18% (37 positive well/87 total wells) for P1, 50% (4/8) for P2, 40% (156/192) for P3, and 5% (20/191) for P4. The mean sequence length downstream of the IS was 336 ± 230 bp (range, 44–1,024 bp). One sequence from P1 (sequence length 456) matched multiple intergenic regions in the Y chromosome and was excluded for further analysis.

Despite a limited number of distinct IS due to clonal expansions, IS was predominant in genes, around 75% of all ISs identified, except for P3 (100% intergenic, $n = 3$ distinct IS). Integration site patterns suggested differences in clonal expansion by disease type, with oligoclonal or polyclonal profiles in polymyositis cases, while the ATLL case

TABLE 1 Patients' characteristics[a]

|  | P1 | P2 | P3 | P4 |
|---|---|---|---|---|
| Age (years) | 31 | 36 | 49 | 73 |
| Sex at birth | Male | Female | Male | Male |
| HTLV pathology | Polymyositis | HTLV-associated myelopathy | Adult T cell leukemia - lymphoma | Polymyositis |
| Country of birth | Congo | Ivory Coast | Senegal | Haiti |
| HTLV-1 viral load, log [cp/$10^6$ cells] | 3.7 (blood) | 4.7 (CSF) | 4.9 (blood) | 4.4 (blood) |
| Treatment | IVIG, prednisone, methotrex-ate | methylprednisolone | Prednisone, idelalisib | None |
| Clinical response | Partial response | Lost to follow-up | Death | Not available |

[a]CSF: cerebrospinal fluid; IVIG: intravenous immunoglobulin; NA: Not available.

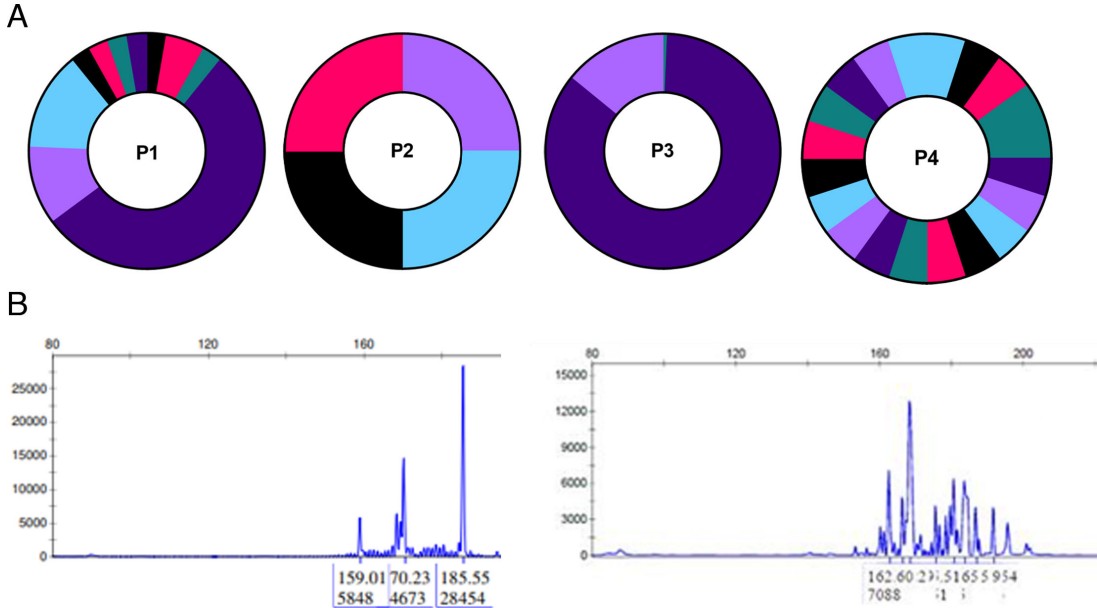

**FIG 1** Clonality of HTLV-I-infected cells in patients with diverse HTLV-I-associated pathology. P1 and P4: polymyositis, P2: HAM (on four sequences only due to the very limited available sample), P3: ATLL. (A) Clonality using the integration site analysis, (B) P1 clonality using TCRgamma (C) P4 clonality using TCR gamma.

exhibited oligoclonal or monoclonal patterns (Fig. 1). Clonality was consistent with TCRγ clonality for P1 and P4, though insufficient sample quantity precluded TCRγ analysis for P2 and P3.

## DISCUSSION

We present an HTLV-1 integration site assay that relies solely on PCR and Sanger sequencing, making it feasible without advanced bioinformatic resources. This assay provided similar results regarding HTLV-1 clonality compared with a TCRγ assay.

Apart from its relatively high yield compared to the 1%–10% of the LM-PCR (9), the primary advantage of this method is its usability without the need for bioinformatic skills. LM-PCR or LAM-PCR assays need complex bioinformatics filtering because of PCR misprimings and chimeric PCR artifacts between LTR and repeated human sequences, an issue that has already led to erroneous conclusions, as experienced by the 2015 *Cell* study by Cohn et al. (9, 19). The use of multiple overlapping primers and independent Sanger sequencing mitigates such issues in our approach. The web tool used in analysis provides an additional safeguard, confirming the immediate proximity of viral and human sequences.

Overall, we present an integration site assay for HTLV-1 that allows an unbiased IS analysis regarding clonal populations in various HTLV-1-associated pathologies.

## ACKNOWLEDGMENTS

This work was supported in part by the Agence Nationale de Recherches sur le SIDA et les hépatites virales-Maladies Infectieuses Emergentes-Medical and Pharmacological ANRS-MIE Network (ANRS-MIE) and the Groupe Pasteur Mutualité (bourse Villa M). Study sponsors had no part in the study design, data collection, data analyses, data interpretation, manuscript writing, nor decision to submit for publication.

V.G. and V.C.: Conceptualization, Methodology, Resources, Formal analysis, Data curation, Writing-original draft; J.A.D. and C.B.: Data curation, Formal analysis, Writing-original draft; S.B.A., E.A., R.L., V.M., C.L., S.C., R.D., M.C., C.M., A.L.J., J.-C.C., and O.B.: Data curation; V.P., A.-G.M., and A.G.-D.: Conceptualization, Writing-review and Editing.

## AUTHOR AFFILIATIONS

[1]Virology Department, INSERM, Pierre Louis Institute of Epidemiology and Public Health (IPLESP), Pitié-Salpêtrière Hospital, Assistance Publique-Hôpitaux de Paris, Sorbonne University, Paris, France

[2]Department of Endocrine and Oncological Biochemistry, Pitié-Salpêtrière Hospital, Assistance Publique-Hôpitaux de Paris, Paris, France

[3]Department of Clinical Hematology, Hôpital Pitié-Salpêtrière, Assistance Publique-Hôpitaux de Paris, Sorbonne University, Paris, France

[4]Department of Neurology, Public Hospital Network of Paris, INSERM, National Center for Scientific Research, Paris Brain Institute, Pitié-Salpêtrière Hospital, Center for Clinical Investigation Neurosciences, Sorbonne University, Paris, France

[5]Department of Internal Medicine and Clinical Immunology, Pitié-Salpêtrière Hospital, Assistance Publique-Hôpitaux de Paris, Sorbonne University, Paris, France

[6]Infectious Diseases Department, Pitié-Salpêtrière Hospital, Assistance Publique-Hôpitaux de Paris, Paris, France

[7]INSERM UMR-S1136, Pierre Louis Institute of Epidemiology and Public Health, Sorbonne Université, Paris, France

[8]Virology Department, Pitié-Salpêtrière Hospital, Assistance Publique-Hôpitaux de Paris, Paris, France

[9]INSERM UMR-S1139, 3PHM, Paris, France

[10]Department of Biological Hematology, Hôpital Pitié-Salpêtrière, Assistance Publique-Hôpitaux de Paris, Sorbonne University, Paris, France

## AUTHOR ORCIDs

Vincent Guiraud (ID) http://orcid.org/0000-0002-4301-3771

## FUNDING

| Funder | Grant(s) | Author(s) |
|---|---|---|
| Agence Nationale de Recherches sur le Sida et les Hépatites Virales | | Vincent Guiraud |
| Groupe Pasteur Mutualité | Villa M | Vincent Guiraud |

## AUTHOR CONTRIBUTIONS

Vincent Guiraud, Conceptualization, Data curation, Formal analysis, Methodology, Supervision, Validation, Writing – original draft | Jérôme Alexandre Denis, Conceptualization, Formal analysis, Methodology, Supervision, Writing – original draft | Sofia Ben Attia, Data curation | Ronan Legrand, Data curation | Véronique Morel, Data curation | Claire Lacan, Data curation | Sylvain Choquet, Formal analysis, Investigation | Rabab Debs, Data curation | Cindy Marques, Data curation | Alexandre Le Joncour, Data curation | Jean-Christophe Corvol, Data curation | Olivier Benveniste, Data curation, Formal analysis | Valérie Pourcher, Data curation | Anne-Geneviève Marcelin, Data curation, Formal analysis, Funding acquisition | Agnès Gautheret-Dejean, Conceptualization, Formal analysis, Methodology, Supervision, Validation, Writing – review and editing | Clotilde Bravetti, Conceptualization, Data curation, Formal analysis, Methodology, Writing – original draft, Writing – review and editing | Vincent Calvez, Conceptualization, Formal analysis, Funding acquisition, Investigation, Methodology, Project administration, Resources, Supervision, Validation, Writing – original draft, Writing – review and editing.

## DATA AVAILABILITY

All integration sites and ddPCR results for quantification are presented as Table S1 and Table S2, respectively. Integration site sequences were deposited under Bioproject PRJNA1231487 in the NCBI database.

## ETHICS APPROVAL

In accordance with the French legislation, this study was registered under number 20241106152758 using the MR-004 referral methodology of the "Commission Nationale de l'Informatique et des Libertés." Patients provided informed consent and were systematically notified of any supplementary biological analyses on frozen samples, initially collected as part of routine clinical practice.

## ADDITIONAL FILES

The following material is available online.

### Supplemental Material

**Fig. S1 (Spectrum03208-24-S0001.docx).** Results of an IS experiment on a 1% agarose gel.
**Supplemental tables (Spectrum03208-24-S0002.xlsx).** Tables S1 and S2.

### Open Peer Review

**PEER REVIEW HISTORY (review-history.pdf).** An accounting of the reviewer comments and feedback.

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
