## [Reviewer comments · Microbiology Spectrum]

Microbiology Spectrum

Development of a new high yield integration site assay reveals disease-specific patterns across HTLV-1-associated pathologies

Vincent Guiraud, Jérôme Denis, Sofia Ben Attia, Erwan Ablin, Ronan Legrand, Veronique Morel, Claire Lacan, Sylvain Choquet, Rabab Debs, Margaux Cheval, Cindy Marques, Alexandre Le Joncour, Jean-Christophe Corvol, Olivier BENVENISTE, Valérie POURCHER, Anne-Geneviève Marcelin, Agnes Gautheret-Dejean, Clotilde Bravetti, and Vincent Calvez

Corresponding Author(s): Vincent Guiraud, Assistance Publique - Hopitaux de Paris

Review Timeline:

Submission Date:	December 9, 2024
Editorial Decision:	January 29, 2025
Revision Received:	February 4, 2025
Editorial Decision:	February 7, 2025
Revision Received:	February 8, 2025
Accepted:	February 20, 2025

Editor: Jérôme Le Goff

Reviewer(s): The reviewers have opted to remain anonymous.

Transaction Report:

DOI: <https://doi.org/10.1128/spectrum.03208-24>

Re: Spectrum03208-24 (Development of a new high yield integration site assay reveals Disease-Specific Patterns across HTLV-1-associated Pathologies)

Dear Dr. Vincent Guiraud:

Thank you for the privilege of reviewing your work. Below you will find my comments, instructions from the Spectrum editorial office, and the reviewer comments.

According to the reviewers' comments and suggestions, we would like you to revise the manuscript in the Observation format of Microbiology Spectrum, which seems more suitable for your work.

Revision Guidelines

Sincerely,
Jérôme Le Goff
Editor
Microbiology Spectrum

Reviewer #1 (Comments for the Author):

This study presents a an HTLV-1 integration site assay that relies only on PCR and Sanger sequencing. Although analyzed sample number was very small, the presented method looks valuable.

Specific comments:

Comment 1: Several methods for analyzing HTLV-1 integration sites have been reported. Comparison of the method presented in this study with these previously-reported methods should be discussed, in particular on the sensitivity.

Comment 2: It is recommended to add results on analysis of more than two uninfected individuals using this method to confirm no existence of false positive data.

Reviewer #2 (Comments for the Author):

Guiraud and co-authors present a short report ("Observations" paper type? - the paper is reviewed assuming this is true) describing a new and improved method to detect integration sites in HTLV-1 infected cells. This is important as integration sites are linked with disease progression, but understanding these has been challenging. This new method adaptation shows promise for future studies to be able to assess integration sites at both a higher throughput and increased depth on a broader scale. This is a straightforward presentation of data and method (in a good way) and I have minimal comments for the authors.

Main comment for the authors:

- Line ~148/line163: how do the authors know that "overall yield was around 20%"? What is the current studies results on integration sites compared to in this regard? Potential integration sites? Previously described integration sites? Other?

Minor comments for the authors:

- A statement on informed consent is missing (approximately line 95). This may be included with the ethics statement (line 139-) but does journal policy require details as to informed consent?
- Clarification question: are the primers listed on lines 109 and 114 (and the subsequent amplification and sequencing primers on lines 124, etc) the same designed in this paper (described in lines 101-105)?
- There are some small grammatical errors (e.g., lines 157, 158, lines 156&157 consistency in Figure/fig, etc) for the authors to correct in the results section prior to proofs though these do not inhibit current review of the paper.
- What does "IS were predominant in genes, around 75% total" lines 153-154 mean? Are 75% of in the integration sites found within genes (vs in intronic regions) identified or were 75% of the integration able to be identified? I'm presuming the former, but a couple of extra words in this sentence may clarify this.

Dear Editor,

We would like to thank the reviewers for their valuable remarks, questions and suggestions. We hope our answers will allow us to improve our manuscript.

Response to the reviewers

Reviewer #1 (Comments for the Author):

This study presents a an HTLV-1 integration site assay that relies only on PCR and Sanger sequencing. Although analyzed sample number was very small, the presented method looks valuable.

We thank the reviewer for his overall positive comment. As both reviewers have highlighted, we sought mostly to present a technical study. This is why we have thoroughly described each step (including PCR ramps) to make this technique easily set up in other laboratories.

Specific comments:

Comment 1: Several methods for analyzing HTLV-1 integration sites have been reported. Comparison of the method presented in this study with these previously-reported methods should be discussed, in particular on the sensitivity.

The question of sensitivity / yield of integration site assay is a key question because it heavily impacts the characteristics of biological samples that can be processed. However, most studies do not disclose either the nature of biological samples (limited blood sampling or apheresis) nor the yield of their integration site (IS) assay. Regarding LM-PCR we have only found one study on HIV that have stated a recovery from 5-15% (with personal communication that a steady 10% yield is an optimistic objective) (1). Regarding the panhandle PCR, one of the reference paper suggested a yield around 9% (2). Interestingly, when we contacted the latter team (which have since also performed LM-PCR (3)), they were less optimistic regarding LM-PCR yields, compared to their panhandle PCR (personal communications). In our experience, we were far below the optimistic 10% yield for LM-PCR, which explains why we focused on a panhandle PCR one.

Regarding sensitivity / limit of detection, we had no trouble if there were few integration sites in a lot of genomic DNA, with inputs up to 1 µg DNA. LM-PCR experiments use up to 1 µg too (there is PCR inhibition above this limit) and can also retrieve several IS, but unfortunately, we have not managed to find a limit of detection in any paper we have red.

Comment 2: It is recommended to add results on analysis of more than two uninfected individuals using this method to confirm no existence of false positive data.

We have systematically used HTLV-negative human DNA as a negative control and had no false-reactive result (no amplicon band), but sometimes a negative result can present as a smear (mostly for experiments with high DNA input), which can be confusing if the agarose gel has not migrated enough. As a consequence, we have added this recommendation in the method part of our manuscript (line 142). We also have added an agarose gel picture of an IS experiment to illustrate the different results.

Reviewer #2 (Comments for the Author):

Guiraud and co-authors present a short report ("Observations" paper type? - the paper is reviewed assuming this is true) describing a new and improved method to detect integration sites in HTLV-1 infected cells. This is important as integration sites are linked with disease progression, but understanding these has been challenging. This new method adaptation shows promise for future studies to be able to assess integration sites at both a higher throughput and increased depth on a broader scale. This is a straightforward presentation of data and method (in a good way) and I have minimal comments for the authors.

We thank the reviewer for his overall positive comment. As both reviewers have highlighted, we sought mostly to present a technical study.

Main comment for the authors:

- Line ~148/line163: how do the authors know that "overall yield was around 20%"?

We simply divided the total number of integration sites we retrieved (after Sanger sequencing) with the total number of integration sites we put as an input, with the latter assessed using digital droplet PCR to get a precise quantification. As there were noticeable variations between all four patients, we also provided distinct results for each patient.

What is the current studies results on integration sites compared to in this regard?

As developed in our answer to the reviewer 1, this point is unfortunately extremely rarely described. Some studies that argue for yields up to 10%, although personal communications with distinct labs that perform IS analyses find these numbers often optimistic, with widely-variable yields between experiments.

Potential integration sites? Previously described integration sites? Other?

Regarding HTLV-integration landscapes, our study was not powered to rigorously compare our results with previously described integration sites, mostly because most of our patients had oligoclonal HTLV proliferations. As a consequence, we had few distinct HTLV-IS to compare genetic or epigenetic features with previous results. However, clonal patterns mirrored previous results, as described in the discussion part of our manuscript.

The most important point, however, is that we also confirmed clonal ratios using an independent assay (the TCR one). This point is of interest since several outdated IS assays such as inverse-PCR (4,5) or transposase-based LM-PCR (6,7) are known to have important bias, due to the relative distance of IS from restriction sites or consensus sequences, respectively.

Minor comments for the authors:

- A statement on informed consent is missing (approximately line 95). This may be included with the ethics statement (line 139-) but does journal policy require details as to informed consent?

Patients provided informed consent and were systematically notified of any supplementary biological analyses on frozen samples, initially collected as part of routine clinical practice. We have corrected this omission.

- Clarification question: are the primers listed on lines 109 and 114 (and the subsequent amplification and sequencing primers on lines 124, etc) the same designed in this paper (described in lines 101-105)?

Yes. We described how we designed the integration primers. We thought it could be useful to explain how they were made in case someone would like to adapt this assay on other retroviruses / gene-therapy vectors.

- There are some small grammatical errors (e.g., lines 157, 158, lines 156&157 consistency in Figure/fig, etc) for the authors to correct in the results section prior to proofs though these do not inhibit current review of the paper.

We thank the reviewer 2 for his thorough reading of our manuscript. We corrected accordingly.

- What does "IS were predominant in genes, around 75% total" lines 153-154 mean? Are 75% of in the integration sites found within genes (vs in intronic regions) identified or were 75% of the integration able to be identified? I'm presuming the former, but a couple of extra words in this sentence may clarify this.

75% of integration sites able to be identified. We have clarified this point in the result part of our manuscript.

References :

1. Wells DW, Guo S, Shao W, Bale MJ, Coffin JM, Hughes SH, et al. An analytical pipeline for identifying and mapping the integration sites of HIV and other retroviruses. *BMC Genomics*. déc 2020;21(1):216.
2. Wagner TA, McLaughlin S, Garg K, Cheung CYK, Larsen BB, Styrchak S, et al. Proliferation of cells with HIV integrated into cancer genes contributes to persistent infection. *Science*. août 2014;345(6196):570-3.
3. Einkauf KB, Lee GQ, Gao C, Sharaf R, Sun X, Hua S, et al. Intact HIV-1 proviruses accumulate at distinct chromosomal positions during prolonged antiretroviral therapy. *Journal of Clinical Investigation*. 28 janv 2019;129(3):988-98.
4. Brady T, Roth SL, Malani N, Wang GP, Berry CC, Leboulch P, et al. A method to sequence and quantify DNA integration for monitoring outcome in gene therapy. *Nucleic Acids Research*. juin 2011;39(11):e72-e72.
5. Oliynyk RT, Church GM. Efficient modification and preparation of circular DNA for expression in cell culture. *Commun Biol*. 21 déc 2022;5(1):1393.
6. Kim J, Park M, Baek G, Kim JI, Kwon E, Kang BC, et al. Tagmentation-based analysis reveals the clonal behavior of CAR-T cells in association with lentivector integration sites. *Molecular Therapy - Oncolytics*. sept 2023;30:1-13.
7. Green B, Bouchier C, Fairhead C, Craig NL, Cormack BP. Insertion site preference of Mu, Tn5, and Tn7 transposons. *Mobile DNA*. déc 2012;3(1):3.

Re: Spectrum03208-24R1 (Development of a new high yield integration site assay reveals Disease-Specific Patterns across HTLV-1-associated Pathologies)

Dear Dr. Vincent Guiraud:

We thank you for your responses to the reviewers and the revisions made to the manuscript. However, you have not addressed the request to format the manuscript according to the "Observation" format, which limits the total word count to 1,200. Before submitting your corrections to the reviewers, could you review the manuscript to ensure it is in the correct format? Details regarding manuscript types can be found at <https://journals.asm.org/journal/spectrum/article-types>.

Revision Guidelines

Sincerely,
Jérôme Le Goff
Editor
Microbiology Spectrum

Dear Editor,

We would like to thank the reviewers for their valuable remarks, questions and suggestions. We hope our answers will allow us to improve our manuscript.

Response to the reviewers

Reviewer #1 (Comments for the Author):

This study presents a an HTLV-1 integration site assay that relies only on PCR and Sanger sequencing. Although analyzed sample number was very small, the presented method looks valuable.

We thank the reviewer for his overall positive comment. As both reviewers have highlighted, we sought mostly to present a technical study. This is why we have thoroughly described each step (including PCR ramps) to make this technique easily set up in other laboratories.

Specific comments:

Comment 1: Several methods for analyzing HTLV-1 integration sites have been reported. Comparison of the method presented in this study with these previously-reported methods should be discussed, in particular on the sensitivity.

The question of sensitivity / yield of integration site assay is a key question because it heavily impacts the characteristics of biological samples that can be processed. However, most studies do not disclose either the nature of biological samples (limited blood sampling or apheresis) nor the yield of their integration site (IS) assay. Regarding LM-PCR we have only found one study on HIV that have stated a recovery from 5-15% (with personal communication that a steady 10% yield is an optimistic objective) (1). Regarding the panhandle PCR, one of the reference paper suggested a yield around 9% (2). Interestingly, when we contacted the latter team (which have since also performed LM-PCR (3)), they were less optimistic regarding LM-PCR yields, compared to their panhandle PCR (personal communications). In our experience, we were far below the optimistic 10% yield for LM-PCR, which explains why we focused on a panhandle PCR one.

Regarding sensitivity / limit of detection, we had no trouble if there were few integration sites in a lot of genomic DNA, with inputs up to 1 µg DNA. LM-PCR experiments use up to 1 µg too (there is PCR inhibition above this limit) and can also retrieve several IS, but unfortunately, we have not managed to find a limit of detection in any paper we have red.

Comment 2: It is recommended to add results on analysis of more than two uninfected individuals using this method to confirm no existence of false positive data.

We have systematically used HTLV-negative human DNA as a negative control and had no false-reactive result (no amplicon band), but sometimes a negative result can present as a smear (mostly for experiments with high DNA input), which can be confusing if the agarose gel has not migrated enough. As a consequence, we have added this recommendation in the method part of our manuscript (line 142). We also have added an agarose gel picture of an IS experiment to illustrate the different results.

Reviewer #2 (Comments for the Author):

Guiraud and co-authors present a short report ("Observations" paper type? - the paper is reviewed assuming this is true) describing a new and improved method to detect integration sites in HTLV-1 infected cells. This is important as integration sites are linked with disease progression, but understanding these has been challenging. This new method adaptation shows promise for future studies to be able to assess integration sites at both a higher throughput and increased depth on a broader scale. This is a straightforward presentation of data and method (in a good way) and I have minimal comments for the authors.

We thank the reviewer for his overall positive comment. As both reviewers have highlighted, we sought mostly to present a technical study.

Main comment for the authors:

- Line ~148/line163: how do the authors know that "overall yield was around 20%"?

We simply divided the total number of integration sites we retrieved (after Sanger sequencing) with the total number of integration sites we put as an input, with the latter assessed using digital droplet PCR to get a precise quantification. As there were noticeable variations between all four patients, we also provided distinct results for each patient.

What is the current studies results on integration sites compared to in this regard?

As developed in our answer to the reviewer 1, this point is unfortunately extremely rarely described. Some studies that argue for yields up to 10%, although personal communications with distinct labs that perform IS analyses find these numbers often optimistic, with widely-variable yields between experiments.

Potential integration sites? Previously described integration sites? Other?

Regarding HTLV-integration landscapes, our study was not powered to rigorously compare our results with previously described integration sites, mostly because most of our patients had oligoclonal HTLV proliferations. As a consequence, we had few distinct HTLV-IS to compare genetic or epigenetic features with previous results. However, clonal patterns mirrored previous results, as described in the discussion part of our manuscript.

The most important point, however, is that we also confirmed clonal ratios using an independent assay (the TCR one). This point is of interest since several outdated IS assays such as inverse-PCR (4,5) or transposase-based LM-PCR (6,7) are known to have important bias, due to the relative distance of IS from restriction sites or consensus sequences, respectively.

Minor comments for the authors:

- A statement on informed consent is missing (approximately line 95). This may be included with the ethics statement (line 139-) but does journal policy require details as to informed consent?

Patients provided informed consent and were systematically notified of any supplementary biological analyses on frozen samples, initially collected as part of routine clinical practice. We have corrected this omission.

- Clarification question: are the primers listed on lines 109 and 114 (and the subsequent amplification and sequencing primers on lines 124, etc) the same designed in this paper (described in lines 101-105)?

Yes. We described how we designed the integration primers. We thought it could be useful to explain how they were made in case someone would like to adapt this assay on other retroviruses / gene-therapy vectors.

- There are some small grammatical errors (e.g., lines 157, 158, lines 156&157 consistency in Figure/fig, etc) for the authors to correct in the results section prior to proofs though these do not inhibit current review of the paper.

We thank the reviewer 2 for his thorough reading of our manuscript. We corrected accordingly.

- What does "IS were predominant in genes, around 75% total" lines 153-154 mean? Are 75% of in the integration sites found within genes (vs in intronic regions) identified or were 75% of the integration able to be identified? I'm presuming the former, but a couple of extra words in this sentence may clarify this.

75% of integration sites able to be identified. We have clarified this point in the result part of our manuscript.

References :

1. Wells DW, Guo S, Shao W, Bale MJ, Coffin JM, Hughes SH, et al. An analytical pipeline for identifying and mapping the integration sites of HIV and other retroviruses. *BMC Genomics*. déc 2020;21(1):216.
2. Wagner TA, McLaughlin S, Garg K, Cheung CYK, Larsen BB, Styrchak S, et al. Proliferation of cells with HIV integrated into cancer genes contributes to persistent infection. *Science*. août 2014;345(6196):570-3.
3. Einkauf KB, Lee GQ, Gao C, Sharaf R, Sun X, Hua S, et al. Intact HIV-1 proviruses accumulate at distinct chromosomal positions during prolonged antiretroviral therapy. *Journal of Clinical Investigation*. 28 janv 2019;129(3):988-98.
4. Brady T, Roth SL, Malani N, Wang GP, Berry CC, Leboulch P, et al. A method to sequence and quantify DNA integration for monitoring outcome in gene therapy. *Nucleic Acids Research*. juin 2011;39(11):e72-e72.
5. Oliynyk RT, Church GM. Efficient modification and preparation of circular DNA for expression in cell culture. *Commun Biol*. 21 déc 2022;5(1):1393.
6. Kim J, Park M, Baek G, Kim JI, Kwon E, Kang BC, et al. Tagmentation-based analysis reveals the clonal behavior of CAR-T cells in association with lentivector integration sites. *Molecular Therapy - Oncolytics*. sept 2023;30:1-13.
7. Green B, Bouchier C, Fairhead C, Craig NL, Cormack BP. Insertion site preference of Mu, Tn5, and Tn7 transposons. *Mobile DNA*. déc 2012;3(1):3.

Re: Spectrum03208-24R2 (Development of a new high yield integration site assay reveals Disease-Specific Patterns across HTLV-1-associated Pathologies)

Dear Dr. Vincent Guiraud:

Your manuscript has been accepted, and I am forwarding it to the ASM production staff for publication. Your paper will first be checked to make sure all elements meet the technical requirements. ASM staff will contact you if anything needs to be revised before copyediting and production can begin. Otherwise, you will be notified when your proofs are ready to be viewed.

Sincerely,
Jérôme Le Goff
Editor
Microbiology Spectrum